# Sound Insulation Performance of Furfuryl Alcohol-Modified Poplar Veneer Used in Functional Plywood

**DOI:** 10.3390/ma15186187

**Published:** 2022-09-06

**Authors:** Shuangshuang Wu, Wei Xu

**Affiliations:** 1Co-Innovation Center of Efficient Processing and Utilization of Forest Resources, Nanjing Forestry University, Nanjing 210037, China; 2College of Furnishings and Industrial Design, Nanjing Forestry University, Nanjing 210037, China

**Keywords:** furfuryl alcohol, poplar veneer, sound insulation, vacuum impregnation

## Abstract

Plywood has poor sound insulation due to its insufficient areal density, which cannot satisfy the demands of an indoor acoustic environment. This report proposed to use furfuryl alcohol to impregnate poplar veneer as a raw material for plywood and explored the sound insulation potential of furfuryl alcohol-modified poplar veneer. The effect of different formulations on the sound insulation performance of modified veneers was discussed, such as furfuryl alcohol concentrations, catalyst categories, and solvent categories. The weight percent gain (WPG) and areal density (AD) were used to evaluate the impregnation effectiveness of furfuryl alcohol modification. The sound insulation was measured by the impedance tube method. The results showed that the WPG of the furfuryl alcohol-modified veneers was evident, and the AD was effectively improved. Furthermore, the average sound insulation of furfuryl alcohol-modified poplar veneer was 25.68~40.10 dB, which increased by 10.8~19.1% compared with that of unmodified veneer. The modified veneer with 50% furfuryl alcohol concentration, taking isopropanol as a solvent, and maleic anhydride as a catalyst, had the optimal sound insulation performance. At the same time, the cell microstructure and chemical components were characterized by scanning electron microscope (SEM), Fourier-transform infrared spectroscopy (FTIR), and Brunauer–Emmett–Teller (BET) theory to explain the sound insulation mechanism further. The results showed that the distortion of cell walls was improved, suggesting a change in the mechanical properties of the cell wall. At the same time, more micropores formed since the filling of furfuryl alcohol resin, yielding a tortuous propagation pathway, so the sound insulation performance improved. Finally, it demonstrated the potential of furfuryl alcohol-modified poplar veneer as raw material to prepare plywood with excellent sound insulation.

## 1. Introduction

With the development of modern industry and traffic, noise has become one of the most dangerous environmental pollution problems, which could cause hearing difficulties, hypertension, and sleeping disorders. There are two main ways to reduce the damage caused by noise. The first is to take action against the noise sources, making facilities produce less noise. Although this method is effective, it is hard for the construction industry to control all noise sources. The second way is to set up barriers with sound insulation materials in the building envelope system to diminish or eliminate sound waves during propagation [1]. However, the soundproofing of envelope systems depends on the quality of sound insulation materials.

Light-weight wood materials, which are beneficial in reducing energy consumption, have been an exciting and innovative alternative in construction systems. Among them, plywood is a long-standing wood product with various applications in furniture [2,3] and interior renovation [4]. However, due to its low density, the sound insulation performance of plywood is not ideal and cannot meet the standards of sound insulation in buildings [5]. For this reason, several studies have focused on developing functional plywood with high sound insulation performance. The traditional method is to increase plywood thickness; however, the enhancement of sound insulation performance is not ideal, wasting many wood resources [6]. Thus, it is gradually replaced by compounding with different tree species or highly damping materials. Sandwich structure plywood with cork as the core layer is widely used in soundproof floors due to cork’s honeycomb structure. The cork layer can effectively buffer when pressed by an external load, thereby increasing the internal friction loss and improving the sound insulation. However, this type of plywood does not have good mechanical properties [7,8]. Considering the excellent mechanical strength of plywood made of eucalyptus [9], Liu et al. (2019) [10] prepared composite plywood for compartment floor with poplar veneer, eucalyptus veneer, and carbon fiber fabric. The composite plywood had a low density of less than 0.7 g·cm^−3^ but had a stable sound insulation performance in the high-frequency band. Ghofrani et al. (2016) used beech and alder with a thickness of 1.8 mm to prepare plywood and used methylene diphenyl diisocyanate (MDI) to compound the rubber layer. The results showed that increasing the beech wood layers and rubber content can improve the acoustic properties of plywood [11]. Some agricultural wastes are also used to improve the sound insulation performance of plywood, such as rice husks [12,13,14]. Furthermore, perforated plywood, as a technical means based on Helmholtz resonators, is also gradually receiving attention [15,16].

Impregnation modification can improve basic wood properties, such as mechanical properties [17], thermal conductivity [18], and acoustic vibration properties [19]. It can also endow wood with special functions, such as transparent wood [20] and conductive wood [21]. It has been considered one of the essential methods to alleviate the shortage of high-quality large-diameter timber and to promote the application and recognition of fast-growing wood. Compared with the laminated method of damping material, the impregnation modification can obtain good sound insulation and ensure that the material is thin and light. Therefore, organic, inorganic, or metal fillers, such as polyurethane [22], calcium carbonate [23], etc., have been used to improve the sound insulation performance of wood materials in recent years by impregnation. These fillers could create a new interface in the internal structure to convert more sound energy into frictional heat during propagation. Thus, the sound insulation performance could be improved. However, such sound insulation function plywood faces problems such as high development cost and unpredictable filler efficacy. Hence, finding an economical and efficient filler for impregnation modification is vital.

Furfuryl alcohol (FA) is prepared by catalytic hydrogenation of furfural gas or liquid phase, and furfural can be obtained from renewable resources such as corn cobs, cottonseed husks, and bagasse. Therefore, wood modified by FA is environment-friendly and has recently received extensive attention [24]. Previous researches have demonstrated that furfurylated wood has better dimensional stability than untreated wood by measuring the shrinkage level. This phenomenon can be explained by the fact that furfuryl alcohol molecules penetrate the wood cells, reducing the number of free pores that the water can enter [25,26]. Moreover, furfurylated wood has greater hardness and elasticity than untreated wood [27]. With the development of wood structure buildings and outdoor furniture, the water resistance, mildew resistance, and termite resistance properties of furfurylated wood have also attracted attention [28,29]. At the same time, some experts pointed out that the final product of furfuryl alcohol-modified wood was not a typical reacted cell wall but more like an ordinary polymer-filled cell lumen [30]. Furfuryl alcohol concentration, catalyst, and solvent have been considered the key parameters affecting the performance of furfurylated wood, but the mechanism of action of furfuryl alcohol-modified wood is still unclear [31]. Compared with common sound insulation fillers, FA has a small molecular weight, low viscosity, and good water solubility, making it convenient to enter the wood. However, little attention was paid to acoustic properties, especially sound insulation performance. Kang et al. (2015) [32] pointed out that FA treatment reduced the natural frequency, static elastic modulus, and dynamic elastic modulus. Based on the correlation between sound and vibration, they inferred that furfurylated wood had the potential to prepare sound insulation floors. However, the sound insulation performance measurement was not carried out in that study, leading to controversy in their conclusion. Miao et al. (2018) [33] explored the acoustic vibration properties of furfurylated wood for instrument soundboards. Moreover, they found that the higher the concentration of FA was, the more significant the negative impact of the vibration acceleration would be. It meant that FA modification could increase the displacement between wood molecules and interfaces, thereby increasing the heat consumed by internal friction. Although the result was not ideal for musical instrument application, it has investigated the potential of furfurylated wood application in sound insulation areas.

In our investigation, taking poplar veneer commonly used in plywood as an object, WPG and AD were discussed to evaluate FA modification’s effectiveness. Then, the effects of FA contents, catalyst categories, and solvent categories on the sound insulation performance were emphatically studied. Apart from those, characterization analysis techniques were used to discuss the sound insulation mechanism of the FA-modified veneer.

## 2. Materials and Methods

### 2.1. Poplar Veneer Specimen Preparation

Poplar (*Populus* spp.) was harvested in Heze, Shandong Province, China. The log was rotary-cut into veneers that were 1.8 mm by 140 mm in cross-section and 140 mm in the longitudinal direction, with the initial moisture content of 13.8%. All the veneers were flat without knots or visible cracks. After being absolutely dried, the veneers were weighed and underwent complete cell impregnation with the furfuryl alcohol (FA) solution. The veneers were presoaked for 30 min to ensure all the veneers were submerged in the impregnation solution, then the vacuum impregnation began. All the veneers were placed in a vacuum drier (PC-3, Shanghai Sanshe Industry Co., Ltd., Shanghai, China) at 14 psi for 30 min, then released the vacuum so that the wood absorbed the FA solution. Finally, it was immersed under normal pressure for 3 h [34,35]. Next, the veneers were wrapped in tin foil and heated in a drying oven (DHG-9643BS-Ⅲ, CIMO, Shanghai, China) at 105 °C for 3 h. Afterward, the tinfoil was removed, and the veneers were heated again at 60 °C for 2 h. Finally, they were dried at 103 °C until being absolutely dried, and the consistent weight was recorded (Figure 1).

Specimens were impregnated with different FA solutions, as listed in Table 1, to explore the influence of FA solution formulation on sound insulation.

Unmodified veneers (RV) were also included as the control, and 10 samples were prepared for each treatment combination. The specimens for the sound insulation test, whose diameters were 100 mm and 30 mm, respectively, were processed by a CNC engraving machine (SC60M, Xunchuan, Shanghai, China).

### 2.2. Evaluation of FA Modification

As a simple and intuitive method, weight percent gain (WPG) was used to reflect the degree of impregnation. It was calculated using Equation (1) by a weighting method.
(1)$$\mathrm{WPG}=\frac{m_{1}-m_{0}}{m_{0}}\times 100\%$$
where *m*_0_ is the absolute dry weight of unmodified veneer, g; *m*_1_ is the absolute dry weight of the veneer impregnated with FA, g.

The sound insulation performance was mainly affected by the mass per unit area and frequency, so areal density (AD) was also used as a parameter for evaluation. It was calculated using Equation (2).
(2)$$\mathrm{AD}=\frac{m}{S}$$
where AD is the areal density, kg·m^−2^; *m* is the absolute dry weight of the specimens, kg; *S* is the area of the incident surface of the sound, m^2^.

### 2.3. Sound Insulation Characteristics

The sound insulation performance of the specimens was carried out by four-microphone impedance tube method according to international standard ISO 12999-2:2020 [36]. The experimental devices were SW422 and SW447 impedance tubes (BSWA TECH, Beijing, China). The inner diameter of SW422 was 100 mm, and that of SW447 was 30 mm.

As shown in Figure 2, the loudspeaker approximately emitted a plane sound wave. It was assumed that the forward wave and reflected wave in the sounding tube were *P*_a_ and *P*_b_, respectively. Similarly, the forward and reflected waves in the receiving tube were *P*_c_ and *P*_d_, respectively. The distances between the four microphones and the specimen’s surface were *x*_1_, *x*_2_, *x*_3_, and *x*_4_, respectively. The sound pressures of the four measuring points were *P*_1_, *P*_2_, *P*_3_, and *P*_4_, respectively [37]. According to the four-channel method, the complex sound pressure could be obtained as described in Equations (3)–(6)
(3)$$P_{1}{=P}_{a}e^{{\mathrm{jkx}}_{1}}{+P}_{b}e^{-{\mathrm{jkx}}_{1}}$$
(4)$$P_{2}{=P}_{a}e^{{\mathrm{jkx}}_{2}}{+P}_{b}e^{-{\mathrm{jkx}}_{2}}$$
(5)$$P_{3}{=P}_{c}e^{{\mathrm{jkx}}_{3}}{+P}_{d}e^{-{\mathrm{jkx}}_{3}}$$
(6)$$P_{4}{=P}_{c}e^{{\mathrm{jkx}}_{4}}{+P}_{d}e^{-{\mathrm{jkx}}_{4}}$$
where *k* is wave number, $k=\frac{2\pi f}{c}$; *f*—frequency, Hz; *c* is the sound speed in the cube, m·s^−1^; *T* is the air temperature, K.

The transmittance coefficient of the specimen could be obtained by Equation (7).
(7)$$\tau_{p}=\frac{P_{c}}{P_{a}}=\frac{{\sin k(x}_{1}-x_{2})}{{\sin k(x}_{4}-x_{3})}\cdot \frac{P_{3}e^{{j\omega x}_{4}}-P_{4}e^{{j\omega x}_{3}}}{P_{1}e^{-{j\omega x}_{2}}-P_{2}e^{-{j\omega x}_{1}}}$$

Finally, the transmission loss of the specimen was found by Equation (8).
(8)$$\mathrm{TL}=-{20\mathrm{lg}|\tau }_{p}|$$

The sound transmission loss ranging from 63 Hz to 1600 Hz was measured with the SW422 impedance tube. Specifically, the measurable frequency range was from 63 Hz to 500 Hz when connecting 0, 2, 3, and 9 microphones. If accessing 1, 2, 3, and 4 microphones, the sound transmission loss in 400~1600 Hz could be measured. Furthermore, the sound transmission loss ranging from 800 Hz to 6300 Hz was measured with the SW477 impedance tube by accessing 5, 6, 7, and 8 microphones. The experiment tested the transmission loss from 100 Hz to 6000 Hz, and the arithmetic average of the transmission loss ranging from 125 Hz to 4000 Hz was used as the average sound insulation [38].

### 2.4. Microscopic Characterization

To observe the distribution of FA in cell walls and lumen, a scanning electron microscope (SEM) was applied using FEI equipment (Quanta 200, MA, USA). Fourier transform infrared (FTIR) Spectrometer (VERTEX 80 V, Bruker, MA, USA) was carried out to find functional molecule changes to explore whether the chemical reaction between FA and wood would affect sound insulation performance. The pore distribution and surface area of the control group (UPV) and the optimal group (OPV) were analyzed by the nitrogen adsorption-desorption method. The samples were both degassed at 80 °C for 10 h in the fully automatic surface and porosity analyzer (Autosorb IQ3, Quantachrome, FL, USA), with the isotherm data analyzed by the BET method and t-plot method.

## 3. Results and Discussion

### 3.1. WPG and AD

Table 2 illustrated the weight percent gain (WPG) and the areal density (AD) of modified veneers impregnated with different FA solution formulations. In the case of a fixed veneer thickness, the trend of areal density (AD) was positively correlated with WPG.

G1, G2, G3, and G4 were veneers impregnated with different FA concentrations. WPG and the growth rate of WPG were increased with FA concentrations. At the same condition of acid catalyst and solvent, FA concentration no longer had a noticeable effect on the pH of the impregnation solution when reaching 30 wt.%.

G3, G5, and G6 were modified veneers using different acid catalysts. G5 treated with oxalic acid (OA) had the highest WPG, followed by maleic anhydride (MA) and citric acid (CA). Different catalysts directly affected the pH of the impregnation solutions, resulting in various degrees of polycondensation of FA. The result revealed that the more acidic environment created by OA in this experiment was more conducive to the polycondensation of FA, thereby increasing the WPG.

G3, G7, and G8 were modified veneers using distilled water (DI), isopropyl alcohol (IPA), and ethanol (EA), respectively. The WPG of FA-modified veneer using IPA as solvent was the best. In previous research, Thygesen et al. (2020) [39] confirmed that the choice of solvent would not affect the reaction product of FA. Therefore, the observation that FA-modified veneer with IPA as the solvent had the highest WPG could be explained by IPA having a lower boiling point than DI and was more volatile during the curing process, restricting the flow of FA, and thus more FA remained in the wood.

### 3.2. Sound Insulation Performance

#### 3.2.1. FA Concentration

Figure 3 gave the transmission loss experiment results of veneers treated with different FA concentrations, taking unmodified veneer as a control. As shown in Figure 3a, both raw poplar veneers and those treated with varying concentrations of FA had poor sound insulation performance at low frequency, mainly in the frequency band below 500 Hz. This phenomenon was understandable because the veneers were primarily affected by stiffness and resonance at low frequency. As a thin plate, the veneer did not have enough stiffness to resist bending deformation when exposed to sound waves, so the sound insulation performance is poor. The results suggested that FA-modified veneers had better sound insulation than unmodified veneers, especially in the middle and high frequency. In addition, due to the coincidence effect, the sound insulation performance of all the samples was sharply reduced at 4000 Hz. However, FA modification could suppress the coincidence valley and improve the sound insulation at the critical frequency.

The average sound insulation improved with the increase of FA concentration until the concentration reached 50% (Figure 3b), consistent with an upward trend of WPG. At 50 wt.% FA concentration, the average sound insulation reached the optimum up to 17.52 dB, 1.47 times higher than the unmodified veneer. However, the average sound insulation decreased while the WPG was still increasing at 70 wt.% FA concentration, indicating other factors affecting the sound insulation performance of the FA-modified veneer. Previous studies have shown that FA concentration affected the penetration of furfuryl alcohol molecules from the cell cavity to the cell wall [40]. In line with these ideas, it could be supposed that the dispersion of FA in the cell lumens might influence sound insulation performance.

#### 3.2.2. Acid Catalyst Category

Figure 4 showed acid catalyst effects on the sound insulation performance. The change of the acid catalyst category did not significantly improve the sound insulation performance at low frequency, either. Below 500 Hz, there was no apparent difference in the sound insulation of the FA-modified veneers, which were catalyzed by maleic anhydride (MA), oxalic acid (OA), and citric acid (CA), respectively. Above 1000 Hz, CA was far less effective than MA and OA in improving sound insulation, even worse than unmodified veneers in some frequency bands (Figure 4a). At 4000 Hz, the sound insulation performance of FA-modified veneers with different catalysts also dropped rapidly.

FA-modified veneers with MA as catalyst had higher average sound insulation (Figure 4b), confirming that MA was an excellent choice for preparation among them. Similarly, the WPG of FA-modified veneer, taking OA as a catalyst, was higher, but the average sound insulation was worse than that taking MA as a catalyst. Previous research showed that hydroxymethyl’s carbocation could react with number b1 carbon on the adjacent furfuryl alcohol furan ring to form furfuryl alcohol resin by self-condensation under acidic conditions. Different acidic conditions would affect the degree of self-condensation [41]. Therefore, we speculated that the degree of self-condensation of furfuryl alcohol might affect the sound insulation performance.

#### 3.2.3. Solvent Category

Figure 5 provided the transmission loss experiment results of FA-modified veneers treated with different solvents. All FA-modified veneers had better sound insulation than unmodified veneers in the test frequency range. Similar to the above two experiments, changing the solvent did not significantly improve the sound insulation performance in the low frequency. Between 1000 and 2000 Hz, IPA was better than the other two solvents, improving the sound insulation performance. Around the critical frequency, IPA and DI were better than EA (Figure 5a). Among the three different solvents, the FA-modified veneer using isopropanol as the solvent had the best average sound insulation, 1.58 times higher than the unmodified veneer (Figure 5b). Moreover, the sound insulation performance of FA-modified veneers with different solvent had a consistent trend as the WPG.

Combined with the results in Table 2, the trend of WPG was consistent with those of AD. It was recognized that the area density was doubled, and the sound insulation performance would be improved by 6 dB. Taking IPA as an example, the sound insulation was increased by 7.2 dB when the areal density was just increased by 0.86 times. This finding supported that the changes in the microstructure and composition of FA-modified veneer might be one of the critical factors affecting the sound insulation performance. Therefore, we further discussed the effects of FA distribution in the wood cell structure and the reaction between FA and wood on acoustic propagation.

### 3.3. SEM Analysis

The morphology change of modified veneers after furfurylation was observed by SEM, as presented in Figure 6.

Figure 6a exhibited the cross-sectional structure of untreated poplar veneer, where cell wall distortion was easily observed. However, the distortion improved or disappeared in other graphs, which showed the cross-sectional structure of FA-modified veneers with different formulations. This phenomenon provided evidence that cell walls were strengthened by FA resin.

Figure 6b–e showed the degree of filling with different FA concentrations. When the FA concentration was below 30%, the vessel and most wood fiber lumens were not filled with FA resin. At the same time, the cell wall thickness did not have a noticeable change, but the cell wall distortion was significantly improved. While the FA concentration was over 50%, most of the vessel and wood fiber lumens were filled, and the cell wall became thicker than usual. The phenomenon was more easily observed in Figure 6e. In the study by Thygesen et al. (2010) [42], the polymer on the cell wall imposed steric constraints on FA so that the FA polymer filled in the lumen had a longer conjugated chain structure than those in the cell wall. Therefore, it was inferred that increasing the FA concentration could help form more polymers with long conjugated structures, controlling other process conditions to be consistent, thereby effectively filling more lumens.

Compared with the control group, the FA-modified veneer with CA as the catalyst improved the distortion of the cell wall, but the effect was not ideal (Figure 6f). Although part of the wood fiber lumens was filled, more cracks appeared in the intercellular layer, which can be explained by that using CA produced a more hydrophobic FA solution [43]. Therefore, FA molecules cannot fill the pores in the cellular structure. As shown in Figure 6g, FA resin appeared in vessel lumens, while it hardly appeared in the wood fiber lumens. This result was due to the lower pH value of FA solution using OA, producing FA polycondensation product with high molecular weight. Compared with Figure 6d, OA fixed more FA resin in the lumens, indicating that OA can promote the polymerization of FA to generate more polymers with long conjugated structures than MA.

As shown in Figure 6h,i, most of the FA resin was fixed in the lumens and cell walls using IPA as a solvent, while FA resin filled only a part of the wood fiber lumens using EA as solvent. It was also judged from the SEM images that the best solvent was IPA, followed by DI and EA, consistent with the WPG trend. Another finding was that EA as the organic solvent resulted in thinner cell walls than samples using water as the solvent, consistent with the reduction in the WPG of FA-modified bamboo using EA in a previous study [44].

### 3.4. FTIR Analysis

The infrared spectra of modified poplar veneers with different FA solutions and the control group were shown in Figure 7, and the characteristic peak assignments were shown in Table 3 [45,46]. It can be seen from Figure 7 that the characteristic peaks of the samples using different catalysts or different FA concentrations were similar. At the same time, the difference in the selected solvent had a greater impact on the absorption peaks. Among all the samples, the characteristic peaks of the FA-modified veneer using IPA as solvent were most similar to those of the unmodified veneer, especially retaining a large number of functional clusters in the fingerprint area (1800~650 cm^−1^).

The modified veneers showed absorption peaks of C-C bonds attributable to a furan ring and branched chain around 786 cm^−1^ [47,48], illustrating that FA resin entered the wood. Furthermore, the effect of FA modification on wood oxygen-containing groups was the most obvious, mainly affecting O-H stretching vibration in lignin and C=O stretching vibration in cellulose, proving that FA could enter the cell wall and complete the solidification. The absorption peaks representing cellulose, hemicellulose, and lignin changed with the control group. The intensity of absorption peaks around 3420 cm^−1^, 2915 cm^−1^, 1740 cm^−1^, and 1050 cm^−1^ decreased significantly, and the absorption peak near 615 cm^−1^ almost disappeared, indicating that wood components were degraded during the FA modification process. This phenomenon may be caused by the cross-linking reaction between FA and wood cell wall components or the thermal and acid conditions during the modification process.

### 3.5. Surface Area and Pore Distribution

Figure 8 showed the pore size distribution of UPV and OPV. As shown in Figure 8a, representing the micropore, OPV showed higher cumulative pore volume than UPV. At the same time, the pore width of OPV was 0.3~0.6 nm, while UPV was 0.6~1.9 mm, as shown in the enlarged drawing. When the pore width ranged from 0.2 nm to 5 nm, OPV showed slightly higher cumulative pore volume than UPV. However, when the pore width was more than 5 nm, OPV had a lower cumulative pore volume than UPV (Figure 8b). This indicated that FA resin filled both the cell lumen and the cell walls, which was consistent with the results in Figure 6h.

Given in Table 4 were the data concerning the surface area and pore volume parameters of UPV and OPV. The FA-modified veneer, denoted as OPV, experienced a slight increase in surface area, which was 1.12 times that of the control sample. Due to the formation of many micropores, OPV had significantly higher total pore volume than UPV. In addition, the average pore size of FA resin-modified poplar veneer was smaller, reaching 2.96 nm, which is reduced by 40%.

### 3.6. FA-Modified Veneer Sound Insulation Mechanism

Figure 9 illustrated the sound propagation of untreated veneers and FA-modified veneers, supported by microstructural results. Firstly, WPG can be significantly obtained by treating poplar veneer with FA, and the worst WPG in the eight formulations also reached 22.41%. The improvement of quality has brought about an increase in areal density, which can improve the sound insulation performance of the material. Fukuta et al. (2012) [49] put forward that the internal structure would affect the material’s sound propagation when exploring the two-component fiberboard’s sound absorption performance. We speculated that a more tortuous and complicated sound pathway was produced when more micropores formed in the poplar veneer. At the same time, FA cured internally, filling the mesopores and macropores in the wood so that the sound waves could not easily penetrate. As a result, more sound energy was converted into heat loss. The G7 with the best impregnation effect had the best sound insulation performance, which was consistent with the inference of this mechanism.

Besides, we also believed that the distribution of FA was also a critical factor affecting sound insulation performance. According to the SEM results, there were three principal distributions of FA resin in the internal structure. One was filled in the cell lumens and the intercellular layer, which was mainly influenced by the degree of polymerization of FA. The second was an attachment to the cell wall, and the third entered the cell wall. Both resulted from the combined effects of the FA polymerization and the cross-linking of FA with chemical components of the cell wall. Although law and order of these three distributions cannot be clearly defined in this work, there was no doubt that various sound media interfaces were formed inside the wood. Li et al. (2016) [50] used nanoindentation technology to explore the modulus and hardness of Masson Pine cell walls treated with different concentrations of FA, using maleic anhydride as a catalyst. The results showed that the modulus and hardness peaked at 50% FA concentration. It was speculated that the pores in the wood structure to accommodate FA resin were limited. Although a higher concentration of FA solution would obtain a larger WPG, more FA resin was retained in the lumens, and excessive penetration may damage the integrity of the cell wall. On this basis, we further inferred that the modulus and hardness of the cell wall could not be effectively improved unless FA resin entered the interior of the cell wall [27]. In this way, more interfaces with different acoustic impedances were formed inside, and the sound tended to reflect into the side with weaker acoustic impedance. The refraction of sound waves increased, consuming more sound energy, thereby improving sound insulation performance. Furthermore, FA impregnation modification might change the wood’s internal acoustic vibration behavior, influencing sound insulation performance. In that case, establishing the cell wall vibration model may be another new way to explore the sound insulation performance of the FA-modified veneer [51].

## 4. Conclusions

It is of great significance to study the sound insulation performance of furfuryl alcohol-modified veneer for enriching the commercial value. It can also provide a theoretical basis for developing sound insulation functional plywood. This work mainly explored the effect of furfuryl alcohol solution formulation on its sound insulation performance and discussed the sound insulation mechanism of furfuryl alcohol-modified poplar veneer. The main conclusions and prospects were as follows:

Furfuryl alcohol modification significantly increased WPG, improving the areal density of poplar veneer. The research proved that furfuryl alcohol modification improved the sound insulation performance of poplar veneer, especially in the human-sensitive frequency band, ranging from 1000 Hz to 3000 Hz. Still, the enhancement effect of furfuryl alcohol modification was not ideal at the low-frequency band.The optimal formulation in this experiment was modified with 50% furfuryl alcohol concentration, using isopropanol as a solvent and maleic anhydride as a catalyst. The preferred modified poplar veneer had average sound insulation of 19.05 dB, 1.6 times higher than the unmodified veneer.Except for the law of mass, the distribution of furfuryl alcohol inside the wood and its crosslinking reaction with wood components might be another critical factor for improving sound insulation performance. From the results of SEM and FTIR, it can be found that furfuryl alcohol modification ameliorated the cell wall distortion, indicating that the mechanical properties of the cell wall were enhanced, resulting in more transmission loss of sound waves inside. The effect of furfuryl alcohol concentration and catalyst on wood components was relatively small, while the components of furfuryl alcohol-modified veneer with different solvents were quite different.The furfuryl alcohol modification technology has been industrialized in Europe [34]. We proved the sound insulation potential of furfuryl alcohol-modified veneers-the primary raw material of plywood, through small batch experiments in the laboratory. Subsequently, batch trial production will be carried out, using furfuryl alcohol-modified veneers to prepare plywood with a larger format. By evaluating the sound insulation performance of the plywood, it can be used in the fields of lightweight partition walls and interior decoration. However, the production cost and target performance must be precisely matched in industrial production. Therefore, it is necessary to systematically study the relationship between furfuryl alcohol resin retention and sound propagation behavior in the future.

## Figures and Tables

**Figure 1 materials-15-06187-f001:**
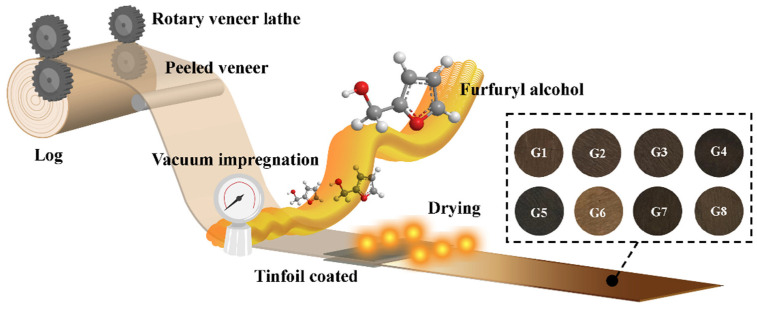
Preparation process of furfuryl alcohol modified poplar veneer.

**Figure 2 materials-15-06187-f002:**
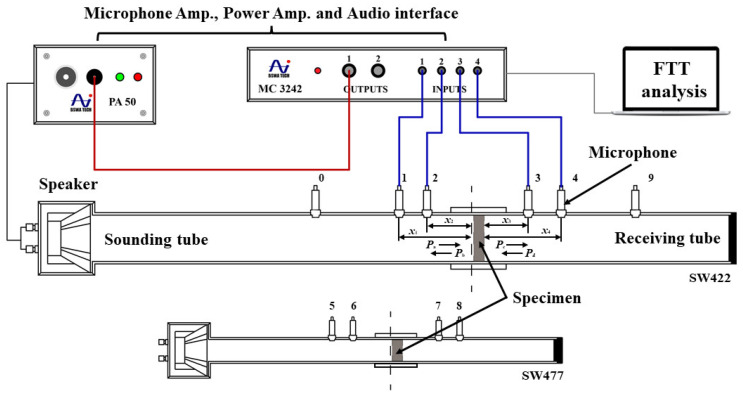
Four-channel sound insulation performance measurement system.

**Figure 3 materials-15-06187-f003:**
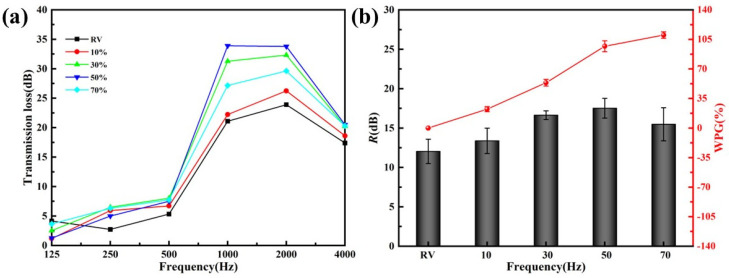
Sound insulation of unmodified veneer and FA-modified veneer with different FA concentrations (**a**) transmission loss dependent on the frequency; (**b**) average sound insulation and WPG.

**Figure 4 materials-15-06187-f004:**
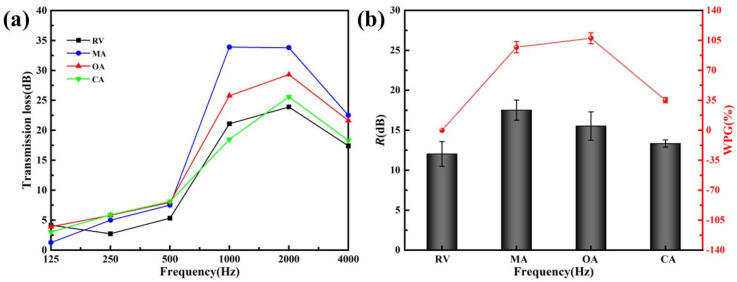
Sound insulation of unmodified veneer and FA-modified veneer with different acid catalysts (**a**) transmission loss dependent on the frequency; (**b**) average sound insulation and WPG.

**Figure 5 materials-15-06187-f005:**
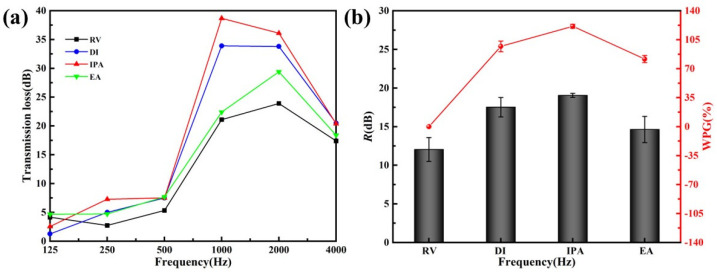
Sound insulation of unmodified veneer and FA-modified veneer with different solvents (**a**) transmission loss dependent on the frequency; (**b**) average sound insulation and WPG.

**Figure 6 materials-15-06187-f006:**
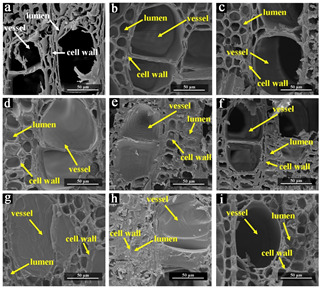
The SEM of unmodified veneer and FA-modified veneer (G1–G8) (**a**) unmodified veneer; (**b**) G1; (**c**) G2; (**d**) G3; (**e**) G4; (**f**) G5; (**g**) G6; (**h**) G7; (**i**) G8.

**Figure 7 materials-15-06187-f007:**
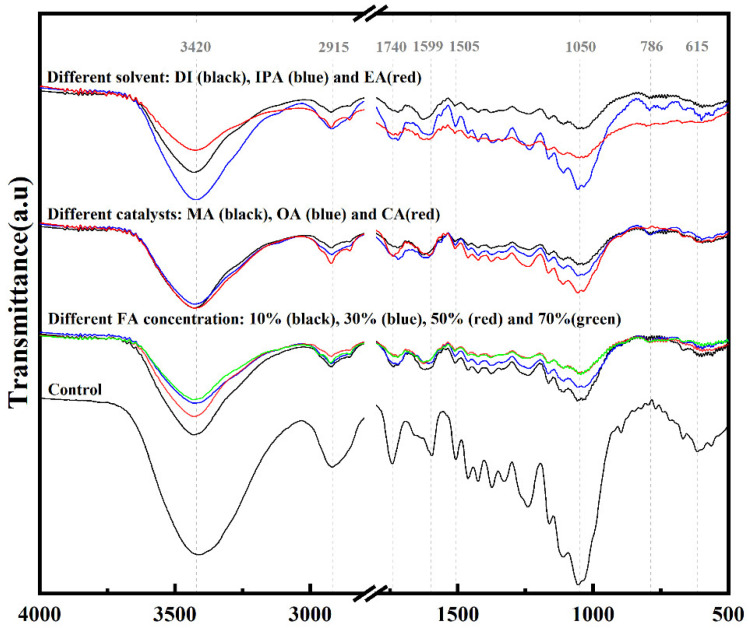
The FTIR of unmodified veneer and FA-modified veneer with different formulations.

**Figure 8 materials-15-06187-f008:**
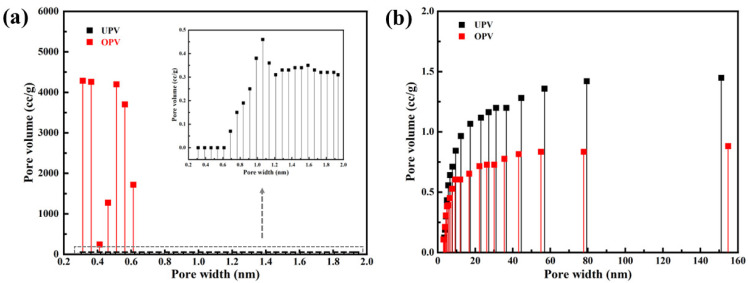
Pore size distribution of UPV and OPV (**a**) micropore; (**b**) mesoporous and megalopore.

**Figure 9 materials-15-06187-f009:**
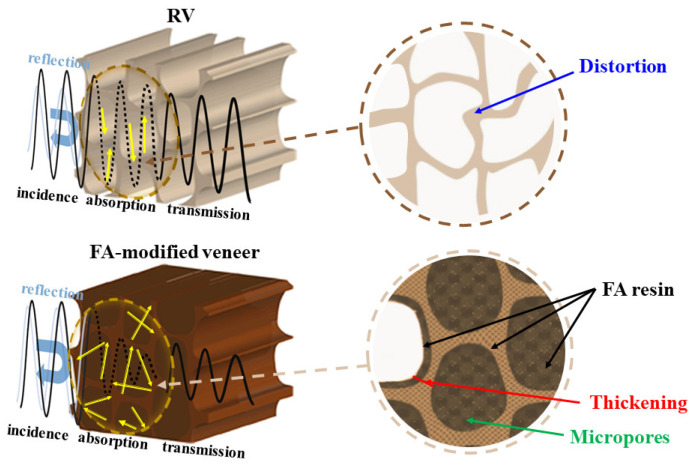
Sound propagation diagram of RV and the FA-modified veneers.

**Table 1 materials-15-06187-t001:** The formulations of the FA solution.

Number	$\omega_{FA}/\mathrm{wt}.\%$	Catalyst	$\omega_{catalyst}/\mathrm{wt}.\%$	$\omega_{Na_{2}B_{4}O_{7}}/\mathrm{wt}.\%$	Solvent	$\omega_{solvent}/\mathrm{wt}.\%$
G1	10	MA	2	2	DI	86
G2	30	MA	2	2	DI	66
G3	50	MA	2	2	DI	46
G4	70	MA	2	2	DI	26
G5	50	OA	2	2	DI	46
G6	50	CA	2	2	DI	46
G7	50	MA	2	2	IPA	46
G8	50	MA	2	2	EA	46

**Table 2 materials-15-06187-t002:** The WPG and AD of unmodified veneer and FA-modified veneers.

Number	pH of FA Solution	WPG/%	AD/(kg·m^−2^)
Control group (RV)	/	0	0.64 ± 0.01
G1	2.5	22.41 ± 2.11	0.72 ± 0.01
G2	3	53.74 ± 3.45	0.86 ± 0.02
G3	3	97.01 ± 6.76	1.05 ± 0.05
G4	3	113.24 ± 3.76	1.15 ± 0.04
G5	2	107.51 ± 6.34	1.06 ± 0.01
G6	3.5	35.08 ± 3.15	0.81 ± 0.04
G7	4	121.28 ± 2.45	1.19 ± 0.04
G8	4.5	81.73 ± 4.31	1.02 ± 0.06

**Table 3 materials-15-06187-t003:** Characteristic absorption peaks and assignments of unmodified veneer and FA-modified veneers with different formulation.

Wavenumber/cm^−1^	Assignments of Absorption Peaks
3420	O-H stretching vibration in lignin
2915	C-H stretching vibration in lignin
1740	Acetyl site of hemicellulose
1595	Stretching vibration of an aromatic skeleton group of lignin
1050	C=O stretching vibration in cellulose
786	Out-of-plane deformation vibration of furan ring -CH bond
615	Out-of-plane deformation vibration of benzene ring -CH bond

**Table 4 materials-15-06187-t004:** Surface area and pore volume parameters of UPV and OPV.

Number	Surface Area/(m^2^·g^−1^)	Total Pore Volume/(cc·g^−1^)	Average Pore Diameter/nm
UPV	3.184	80.98	4.95
OPV	3.562	15836.79	2.96

## Data Availability

The data presented in this study are available on request from the corresponding author.

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
