# Peer review of "Sound Insulation Performance of Furfuryl Alcohol-Modified Poplar Veneer Used in Functional Plywood"

_materials, 2022, doi:10.3390/ma15186187_

Round 1

Reviewer 1 Report

Attached

Author Response

Dear Reviewer:

Dear Reviewer:
I am pleased to resubmit for the revised version of manuscript entitled “Sound insulation performance of furfuryl alcohol-modified poplar veneer used in functional plywood” (ID: 1901724). Thank you for reading our manuscript and reviewing it. Those comments are all valuable and very helpful for revising and
improving our paper. We have revised our manuscript carefully and have made correction which we hope meet with approval. So, we have sent the revised manuscript and have highlighted changes by using the yellow color. The main corrections in the paper and the responds to your comments are as following:

Q1. Line (25): – ‘....... the mechanical properties of the wood cell wall...... ‘was there any mechanical analysis to determine the strength of the cell wall in the study? If no, on what basis that the author(s) mention that the mechanical properties of the wood cell wall were improved?

A: Thanks for your advice, we didn’t characterize the mechanical properties of the cell wall, but we have done this in the following work. In this paper, we put forward this conjecture based on the improvement of the cell wall aberration problem shown in SEM pictures. In order to better understanding, we have revised the expression.
This answer is incorporated in the text on page 1, lines 25-27 in the revised version,thank you very much.

Q2. P3 line 107 and 109-113 – I would suggest the author(s) provide the moisture
content of the veneer before and after treatment to give better understanding to the
reader.

A: Thanks for your advice, we dried the veneer at 103
°C before impregnation and after curing to make sure the veneers was absolutely dry(with a moisture content of 0). We used absolutely dried to explain it. And for better understanding we supplemented the
initial moisture content of poplar veneer.

This answer is incorporated in the text on page 3, line 122 and lines 124-129 in the revised version, thank you very much.

Q3. P5 line 196-197 – ‘...we suspected that IPA had a lower boiling point than DI and was more volatile ......’ it was well known that isopropyl alcohol had a lower boiling point and more volatile compared to water and it was not suspected. Please revise the sentence.

A: Thanks for your advice, we have revised the sentence by that “Therefore, the FA-modified veneer with IPA as the solvent had the highest WPG could be explained by
that IPA had a lower boiling point than DI and was more volatile during the curing process, restricting the flow of FA, and thus more FA remained in the wood”.

This answer is incorporated in the text on page 6, lines 217-220 in the revised version, thank you very much.

Q4. P9 Figure 6 – I would recommend that the author(s) labelled the SEM image, showing the vessel and lumen that have been treated and not treated.

A: Thanks for your advice, we have replaced the picture. For uniformity and better understanding, we renamed each SEM image with the number shown in Table 1.

This answer is incorporated in the text on page 9, line 329 in the revised version, thank you very much.

Q5. P11 line 369-370 – The author(s) reported that the chemical components of poplar wood influenced the sound insulation performances of the material. I would suggest the author(s) to provide the chemical composition of poplar wood to support this statement.

A: Thanks for your advice, we realized that there was an inappropriate expression in this section. In this paragraph, we mainly discussed the effect of FA distribution on the sound insulation performance, so we deleted “the chemical components of poplar wood”and revised this sentence.

This answer is incorporated in the text on page 11, lines 389-391 in the revised version, thank you very much.

Reviewer 2 Report

The manuscript is focused on investigation of the sound insulation properties of poplar veneer, modified with furfuryl alcohol, and intended for manufacturing plywood panels. Overall, the manuscript is well-written, structured and informative, but needs a major revision before its acceptance for publication in Materials Journal. Please, see below my comments on your work:

In general, the abstract of the manuscript (lines 10 to 28) and the keywords (line 29) correspond to the title, aims and objectives of the paper. The abstract is informative, and contains the main findings of the article.

Line 20: I’d recommend to replace “raw” with “untreated/unmodified” veneer.

Line 23: although well known, please add the full terms “scanning electron microscope” and “Fourier-transform infrared spectroscopy”, followed by the common abbreviations SEM and FTIR.

Line 24-25: please revise/replace the sentence “It proposed that the sound insulation mechanism of furfuryl alcohol-modified poplar veneer obeyed the law of mass.”, now it is not very clear.

Line 28: Please replace/delete “available”, it does not make any sense.

Line 45: “…cannot meet the actual need.” – please revise/replace this statement, now it is not very clear.

Please add information on more relevant previous studies on sound insulation performance of plywood.

Line 59: when referring to plywood fabricated from Eucalyptus, please check this relevant paper: https://doi.org/10.1155/2022/8000780

Lines 75-76: Besides eco-friendliness, furfurylated wood has other well-known advantages, such as greater dimensional stability and biological durability, please discuss them as well, and provide additional references to support your statements.

In general, the Introduction part is well prepared, but can be further extended based on the above given comments. I’d recommend the authors to include more relevant references on the research topic.

 Lines 108-109: please provide additional explanation. Why exactly 30 minutes? Please better explain the vacuum impregnation parameters.

 Line 110: please provide relevant information on the vacuum drier used.

 Line 116, Figure 1: please provide a better image, if possible.

Line 138-139: please add the standard to the references of your manuscript.

Overall, the Materials and Methods section is well written and provides relevant information on the materials and methods applied in this research work.

The Results and Discussion is informative, but some parts are not properly discussed with previous research works in the field.

Line 334, Figure 7: please replace with a better quality figure, if possible.

The Conclusion part reflects the main findings of the research. Here I’d recommend to add how the results of your research work can be integrated into industrial manufacturing practice.

The references cited are appropriate and correspond to the topic of the manuscript. However, the addition of more relevant references, especially in the Introduction and Results and Discussion sections, would significantly improve the scientific soundness of the presented manuscript.  

Best regards!

Please add information on more relevant previous studies on sound insulation performance of plywood.

Line 59: when referring to plywood fabricated from Eucalyptus, please check this relevant paper: https://doi.org/10.1155/2022/8000780

Lines 75-76: Besides eco-friendliness, furfurylated wood has other well-known advantages, such as greater dimensional stability and biological durability, please discuss them as well, and provide additional references to support your statements.

In general, the Introduction part is well prepared, but can be further extended based on the above given comments. I’d recommend the authors to include more relevant references on the research topic.

Lines 108-109: please provide additional explanation. Why exactly 30 minutes? Please better explain the vacuum impregnation parameters.

Line 110: please provide relevant information on the vacuum drier used.

Line 116, Figure 1: please provide a better image, if possible.

Line 138-139: please add the standard to the references of your manuscript.

Overall, the Materials and Methods section is well written and provides relevant information on the materials and methods applied in this research work.

The Results and Discussion is informative, but some parts are not properly discussed with previous research works in the field.

Line 334, Figure 7: please replace with a better quality figure, if possible.

The Conclusion part reflects the main findings of the research. Here I’d recommend to add how the results of your research work can be integrated into industrial manufacturing practice.

The references cited are appropriate and correspond to the topic of the manuscript. However, the addition of more relevant references, especially in the Introduction and Results and Discussion sections, would significantly improve the scientific soundness of the presented manuscript.  

Author Response

Dear Reviewer:

I am pleased to resubmit for the revised version of manuscript entitled “Sound insulation performance of furfuryl alcohol-modified poplar veneer used in functional plywood” (ID: 1901724). Thank you for reading our manuscript and reviewing it. Those comments are all valuable and very helpful for revising and improving our paper. We have revised our manuscript carefully and have made correction which we hope meet with approval. So, we have sent the revised manuscript and have highlighted changes by using the yellow color. The main corrections in the paper and the responds to your comments are as following:

Q1. Line 20: I’d recommend to replace “raw” with “untreated/unmodified” veneer.

A: Thanks for your advice, we have revised it and replaced expressions in the full text with “unmodified veneer”.

This answer is incorporated in the text on page 1, line 20 in the revised version, thank you very much.

Q2. Line 23: although well known, please add the full terms “scanning electron microscope” and “Fourier-transform infrared spectroscopy”, followed by the common abbreviations SEM and FTIR.

A: Thanks for your advice, we have added the full terms.

This answer is incorporated in the text on page 1, lines 23-24 in the revised version, thank you very much.

Q3. Line 24-25: please revise/replace the sentence “It proposed that the sound insulation mechanism of furfuryl alcohol-modified poplar veneer obeyed the law of mass.”, now it is not very clear.

A: Thanks for your advice, we have deleted this sentence and revised this part.

This answer is incorporated in the text on page 1, lines 25-27 in the revised version, thank you very much.

Q4. Line 28: Please replace/delete “available”, it does not make any sense.

A: Thanks for your advice, we have delated it.

This answer is incorporated in the text on page 1, line 29 in the revised version, thank you very much.

Q5. Line 45: “…cannot meet the actual need.” – please revise/replace this statement, now it is not very clear.

A: Thanks for your advice, we have revised this statement and added relevant reference.

This answer is incorporated in the text on page 2, line 46 in the revised version, thank you very much.

Q6. Please add information on more relevant previous studies on sound insulation performance of plywood.

A: Thanks for your advice, we have added some information on the relevant previous studies.

This answer is incorporated in the text on page 2, lines 48-66 in the revised version, thank you very much.

Q7. Line 59: when referring to plywood fabricated from Eucalyptus, please check this relevant paper: https://doi.org/10.1155/2022/8000780

A: Thanks for your advice, we have added the reference and revised these sentences.

This answer is incorporated in the text on page 2, lines 55-56 in the revised version, thank you very much.

Q8. Lines 75-76: Besides eco-friendliness, furfurylated wood has other well-known advantages, such as greater dimensional stability and biological durability, please discuss them as well, and provide additional references to support your statements.

A: Thanks for your advice, we have added relevant discussion and references.

This answer is incorporated in the text on page 2, lines 84-100 in the revised version, thank you very much.

Q9. Lines 108-109: please provide additional explanation. Why exactly 30 minutes? Please better explain the vacuum impregnation parameters.

A: Thanks for your advice, the impregnation time was based on previous studies and our pre-experiment. In order to better understanding, we added some information and revised this part.

This answer is incorporated in the text on page 2, lines 123-128 in the revised version, thank you very much.

Q10. Line 110: please provide relevant information on the vacuum drier used.

A: Thanks for your advice, we have added relevant information.

This answer is incorporated in the text on page 3, line 126 in the revised version, thank you very much.

Q11. Line 116, Figure 1: please provide a better image, if possible.

A: Thanks for your advice, we have replaced the picture and provided high-resolution picture in the attachment.

This answer is incorporated in the text on page 3, line 133 in the revised version, thank you very much.

Q12. Line 138-139: please add the standard to the references of your manuscript.

A: Thanks for your advice, we have supplemented the reference.

This answer is incorporated in the text on page 4, lines 156-157 in the revised version, thank you very much.

Q13. Line 334, Figure 7: please replace with a better quality figure, if possible.

A: Thanks for your advice, we have replaced the picture and provided high-resolution picture in the attachment.

This answer is incorporated in the text on page 10, line 352 in the revised version, thank you very much.

Q14.The Conclusion part: I’d recommend to add how the results of your research work can be integrated into industrial manufacturing practice.

A: Thanks for your advice, we have supplemented the advantage of furfuryl alcohol impregnation modification as well as problems in industrial production practice, and presented the outlook.

This answer is incorporated in the text on page 13, lines 440-450 in the revised version, thank you very much.

Round 2

Reviewer 2 Report

The authors have addressed all my comments/remarks in the revised version of the manuscript, and I believe it can be accepted for publication in the current form.